# Performance of Slow-Growing Chickens Fed with *Tenebrio molitor* Larval Meal as a Full Replacement for Soybean Meal

**DOI:** 10.3390/vetsci9030131

**Published:** 2022-03-12

**Authors:** Jaime Nieto, Javier Plaza, Javier Lara, José-Alfonso Abecia, Isabel Revilla, Carlos Palacios

**Affiliations:** 1Area of Animal Production, Faculty of Environmental and Agrarian Sciences, University of Salamanca, Avenida Filiberto Villalobos 119-129, 37007 Salamanca, Spain; pmjavier@usal.es (J.P.); carlospalacios@usal.es (C.P.); 2Animal Husbandry and Animal Health Assistance, High-School Torres Villarroel, Avenida Hilario Goyenechea, 44, 37008 Salamanca, Spain; jlaram@educa.jcyl.es; 3Institute of Research in Environmental Sciences of Aragón (IUCA), University of Zaragoza, Miguel Servet, 177, 50013 Zaragoza, Spain; alf@unizar.es; 4Area of Food Technology, E.P.S. of Zamora, University of Salamanca, Avenida Requejo 33, 49022 Zamora, Spain; irevilla@usal.es

**Keywords:** alternative poultry farming, slow-growth chicken, insect meal, *Tenebrio molitor*, alternative protein sources

## Abstract

Insect larval meal is an increasingly common protein source in poultry systems. In this study, the effect of replacing soybean meal with *Tenebrio molitor* larval meal on the performance of slow-growing chickens was assessed. A total of 128 one-day-old chickens (*Colorield*) were randomly divided into a control group (C) (*n* = 64), fed with soybean meal, and an experimental group (TM) (*n* = 64), fed with *T. molitor* larvae meal. The chicks were slaughtered after 95 days. Three different isoenergetic and isoproteic diets (F1, F2 and F3) were used for each group. The F1 diet resulted in higher body weight gain and higher feed and water intakes in group C, but a lower feed conversion ratio. Contrarily, diets F2 and F3 did not produce differences in the studied parameters between the two groups, except for body weight gain in the case of diet F2, which was highest in group C. Therefore, weight gain and feed and water intakes were significantly higher in group C, but there were no differences in feed conversion ratio or live weight. In conclusion, the total replacement of soybean meal with *T. molitor* larvae meal resulted in a reduction in feed intake and a consequent reduction in weight. During this period, partial rather than total substitution may be recommended.

## 1. Introduction

Global meat consumption is increasing due to the rising incomes and changing dietary patterns of the population. Poultry production is a prime way to meet the demand for animal protein. However, it should become more sustainable regarding the environment and the type of feed used in this sector [1]. Indeed, in modern society, the sustainability of food production and processing systems based on increased efficiency of raw material use, low pollution and reduced waste has become a priority [2].

Soybean meal is the most widely used source of protein in non-ruminant animal feed [3]. However, in recent years, the rising price of this raw ingredient has jeopardized poultry meat production, even more so in some developing countries [4]. The severe environmental problems related to soya production are also noteworthy, as it is one of the main causes of deforestation in the American continent, and in addition, it requires a very high input of water and phytosanitary products [5]. Therefore, the use of non-soybean protein sources in poultry diets is essential to avoiding negative social, economic and ecological impacts arising from large-scale soybean imports. Currently, there is an emerging trend towards the use of alternative feeds to soybeans for poultry which are capable of achieving feed efficiency and feed quality equivalent or superior to soyabean feeds [6,7,8].

A natural behavior of chickens is feeding on different insects throughout their lives [9]. Therefore, it is very likely that chickens would do well with insect meal as part of their dietary rations [10]. Insects are claimed to be an alternative and sustainable nutritional source for farm animals, including poultry [8,11,12]. It is a high-protein feed [13], which also has a considerable amount of unsaturated fats (specifically linoleic and linolenic acids), and large amounts vitamins, fiber and minerals [14,15]. Furthermore, it is worth noting that insects-based diets have gained commercial interest in poultry farming since the recent EU regulation allowing the use of insect meal in monogastric diets [16]. However, to know the level of sensible inclusion for this type of meal in the animal’s diet is essential. Numerous studies have been conducted to elucidate the inclusion percentage of this type of meal that optimizes chicken growth [17,18,19].

Alongside the production sustainability provided by insect meal inclusion, slow-growing chicken strains are being used more often in the meat sector [20]. Due to their growth and better adaptation to alternative production systems, the life cycles of these strains generally exceed 60 days [21,22]. Hence, although they require more time to reach market weight, they may exhibit nutritional and organoleptic characteristics that are more appreciated and demanded by consumers [23]. Moreover, feed conversion of slow-growing chicken can reach even better results when using new raw materials included in their diets [23]. In this context, an increasing number of insect species are used in the production of meal for feed purposes in the livestock sector. In particular, the increase in growth performance in poultry fed with meal of the coleopteran species *Tenebrio molitor*, and the changes that its inclusion generate in the immune system of the animals, have been demonstrated by several authors [18,24,25]. However, further research is needed to determine how such inclusion affects the mechanisms for transforming food into productive yields.

The present study aimed to assess the effect of the total replacement of soybean meal as a protein source with *Tenebrio molitor* larval meal on slow growing chicken development, throughout their complete lifecycle. They were always fed isoenergetic and isoproteic diets.

## 2. Materials and Methods

### 2.1. Animals and Experimental Design

All slow-growing chickens (hereinafter named as chickens) used in the trial were treated in accordance with the principles of European Parliament and Council Directive 2003/65/EC of 22 July 2003, amending Directive 86/609/EEC regarding the protection of animals used for experimental and other scientific purposes. These animals were manipulated under Order ECC/566/2015, of 20 March, establishing the training requirements to be met by personnel handling animals used, bred or supplied for experimental and other scientific purposes, including teaching. The experimental protocol was approved by the Bioethics Committee of the University of Salamanca (Spain) with registration number 590, regulated by RD 53/2013, of 1 February, establishing the basic rules applicable for the protection of animals used in experimentation and other scientific purposes, including teaching. The trial was conducted by personnel from the Animal Production area in the facilities and experimental fields of the University of Salamanca.

A total of 128 one-day-old male slow-growing chickens (*Colorield*) were used in this trial. These animals were previously vaccinated at the hatchery against Marek’s disease and avian infectious bronchitis. Once in the experimental facilities, they were homogeneously divided into two groups: (1) the control group (C) with a mean weight of 39.89 ± 0.04 g that was fed soybean meal as the main protein source, and (2) the tenebrio group (TM) with a mean weight of 39.82 ± 0.10 g that was fed *Tenebrio molitor* larvae meal. Each group consisted of 64 chickens, divided in turn into eight replicates of eight birds each. The animals were raised until 95 days of age, then slaughtered in an authorized slaughterhouse in compliance with Directive 91/628/EEC on good animal welfare during transport and the slaughter procedure in Regulation 93/119/EC.

From their arrival until 50 days of life, each replicate was housed in a 1.50 × 0.60 m pen located in an enclosed housing, provided with a drinking trough and a circular feeder. The stocking rate in each pen was 9.79 chickens/m^2^. First age feeders and drinkers were used for the first 21 days, being replaced later by second age ones. The floor was covered with 6 cm thick wood shavings. Temperature and humidity were controlled to maintain optimum conditions for the animals according to their age. The average temperature inside the building was 32.5 ± 1.4 °C during the first week, 28.6 ± 1.9 °C during the second week, 24.7 ± 2.1 °C during the third week and 21.5 ± 2.4 °C during the rest of the experiment. The chicks received a natural 10 h photoperiod.

From 50 days old until the end of the study, the eight chicken replicates were placed in an outdoor pen with an open-air area, with circular feeders and drinkers, and a shelter for protection, allowing them complete free movement. This shelter had a single entrance on the south side, an opening at the top to ensure ventilation and a translucent section for lighting. The stocking rate in this second stage was 0.25 chickens/m^2^. The outside yard had natural soil, and it was perimetrically enclosed by a 1 m high metal sheet (buried 30 cm) and a 13 × 13 mm grid plastic mesh covering the sides and the roof to prevent predators from entering. The pens were divided with plastic mesh and the access doors were made of untreated wood.

### 2.2. Composition of the Diets and Data Collection

Three different concentrated diets (F1, F2 and F3) were used throughout the process, which were changed every 4 weeks. For each of these three diets, the two experimental groups of chickens (C and TM) were iso-energetically and iso-proteically fed. In particular, F1 was used from day 1 to day 29, F2 from day 29 to day 57 and F3 from day 57 to day 95. Thus, following the nutritional recommendations of the National Research Council (NRC) [26] for medium-slow differentiated growth chickens, we attempted to meet the nutritional needs of the chickens.

The compositions of the three diets, expressed in terms of percentages of raw matter, are shown in Table 1.

The chemical/nutritional characteristics of these diets were previously analyzed (Table 2) in the food analysis laboratories of the Abiomed Higiene S.L. company (Salamanca, Spain), which operates under an ISO 9001 certified quality system.

The entire trial, including both the indoor and outdoor stages, lasted 95 days and was divided into three phases, according to the diets used (Table 1 and Table 2). The diets were supplied in fine grind form. Feed and water were supplied ad libitum. The health status of the animals was monitored daily during the whole experimental period. During the 95-day experiment, the following parameters of the chickens were measured weekly: live weight (LW), feed intake (FI) and water intake (WI), by using a dynamometer and a digital balance. Subsequently, body weight gain (BWG), as an indicator of growth rate, and feed conversion ratio (FCR), which indicates the relationship between feed intake and weight gain of the animal, were calculated.

### 2.3. Statistical Analysis

Statistical processing of the data was performed using IBM-SPSS Package 26 software (IBM, Chicago, IL, USA). Each group of eight chickens (replicate) was considered a sampling unit (*n* = 16). Significant differences between C and TM groups for each of the studied parameters were studied by multivariate analysis of variance (MANOVA) fitted to a general linear model (GLM), taking diet and group as fix factors and the initial weights at the second and third stages as covariants, given that those weights were different for each group. Before carrying out the described statistical tests, the Kolmogorov–Smirnov test was performed to check the normality of the recorded data. The statistical significance was assessed at the 95% confidence level (*α* = 0.05) using Snedecor’s F as the contrast statistic.

Differences in the BWG increment (%) between C and TM groups throughout the trial were analyzed using a Student’s *t*-test for paired samples. In contrast, differences in the BWG increment (%) and in FI increment (%) between the C and TM groups for each of the three diets in the trial (F1, F2 and F3) were analyzed using a Student’s *t*-test for independent samples.

All the results are expressed as means and standard errors of the mean (SEM).

## 3. Results

### 3.1. Productive Performance

Throughout the experiment, the chicks remained healthy, and no deaths were observed in any of the groups. The results during this feeding period (Table 3) show that the C group reached a significantly higher LW (*p* < 0.01) than the TM group. Consequently, the BWG during the first 29 days of life was also significantly higher (*p* < 0.01) in the chickens of group C than in those of group TM. Those results are directly related to those obtained for FI, in which BWG was likewise significantly higher (*p* < 0.01) for the C group than for the TM group. In addition, the chickens in group C consumed significantly more water than those in group TM. Finally, regarding the FCR, significant differences were also found between the two groups (*p* < 0.01), with the highest value coming from the TM group.

Weekly analysis of LW revealed significant differences (*p* < 0.01) between the two groups after 15 days of life (Table 4), and these differences were maintained until the end of this first period. The C group chickens reached the highest weights. These results are in agreement with those shown in Table 3. As for BWG, significant differences (*p* < 0.01) were found between groups in the 2nd, 3rd and 4th weeks. Again, higher values in all cases were reached by group C. The FI was significantly higher for group C during the 1st and 4th weeks (*p* < 0.05, *p* < 0.01, respectively). In contrast, WI showed significant differences (*p* < 0.01) throughout the entire phase, the higher values in all cases being for the C group. For the FCR, the opposite situation to the FI occurred, since significant differences (*p* < 0.01) were obtained between groups in the 2nd and 3rd weeks, the values being higher in both cases in group C.

The results during this diet (Table 3) revealed no differences between the two groups in relation to LW, FI, WI and FCR. Nevertheless, at the end of this period, chickens in group C reached a higher weight and had higher feed and water consumption than those in group TM. Conversely, BWG was significantly higher (*p* < 0.05) in the chickens of group C compared to those of group TM. The weekly results (Table 4) did not show any significant difference between groups in any of the variables studied during this period.

The results for diet F3 (Table 3) did not reveal significant differences in any of the studied variables. However, group C reached higher values for all of them. Weekly analysis (Table 4) showed that group C obtained significantly higher LW values (*p* < 0.05) than group TM at 64, 78 and 92 days of life. Similarly, FI was significantly higher (*p* < 0.05) at the 9th week in the C group than in the TM group, which analogously occurred at the 10th week for WI (*p* < 0.01). By contrast, BWG and FCR did not present differences in any of the weeks of this last period.

**Table 4 vetsci-09-00131-t004:** Weekly production performance parameters of the two studied chicken groups (C and TM) for the three different diets (mean ± SEM).

P	G	Indoor Stage									Outdoor Stage					
		F1 (0–29 d)					F2 (29–57)				F3 (57–95)					
		0 w (0–1 d)	1 w (1–8 d)	2 w (8–15 d)	3 w (15–22 d)	4 w (22–29 d)	5 w (29–36 d)	6 w (36–43 d)	7 w (43–50 d)	8 w (50–57 d)	9 w (57–64 d)	10 w (64–71 d)	11 w (71–78 d)	12 w (78–85 d)	13 w (85–92 d)	14 w (92–95 d)
LW ^1^	C	39.9 ± 0.0	83.8 ± 2.6	191.5 ± 5.1	351.8 ± 11.7	583.6 ± 13.6	949.4 ± 17.2	1368.6 ± 31.7	1772.7 ± 35.0	2229.0 ± 45.3	2756.7 ± 51.6	3237.0 ± 53.2	3493.5 ± 66.5	4125.0 ± 70.0	4598.4 ± 73.5	4739.0 ± 73.5
	TM	39.8 ± 0.1	78.3 ± 2.9	156.9 ± 4.4	268.7 ± 9.2	407.5 ± 14.8	643.6 ± 18.3	956.9 ± 23.8	1267.3 ± 25.5	1643.2 ± 49.2	2047.5 ± 59.5	2408.2 ± 68.9	2803.0 ± 78.2	3403.0 ± 97.7	3841.7 ± 111.7	3981.5 ± 124.4
	Sig.	NS	NS	**	**	**	NS	NS	NS	NS	*	NS	*	NS	*	NS
BWG	C	-	6.3 ± 0.4	15.4 ± 0.6	22.9 ± 1.0	33.1 ± 1.3	52.3 ± 1.0	59.9 ± 3.4	57.7 ± 4.1	67.8 ± 2.2	75.4 ± 2.1	68.6 ± 2.1	36.6 ± 5.5	85.1 ± 2.6	67.6 ± 2.5	46.8 ± 3.5
	TM	-	5.5 ± 0.4	11.2 ± 0.4	16.0 ± 0.9	19.8 ± 1.0	33.7 ± 1.1	44.8 ± 1.8	44.3 ± 2.3	59.1 ± 3.0	57.7 ± 2.5	51.5 ± 2.7	56.4 ± 3.0	85.7 ± 4.6	62.7 ± 4.0	46.6 ± 6.1
	Sig.	-	NS	**	**	**	NS	NS	NS	NS	NS	NS	NS	NS	NS	NS
FI	C	-	11.5 ± 0.9	28.1 ± 1.5	41.1 ± 1.2	70.7 ± 3.8	111.3 ± 2.0	134.9 ± 3.7	122.3 ± 3.7	167.2 ± 0.2	209.1 ± 1.8	233.4 ± 2.1	212.2 ± 0.3	260.8 ± 3.3	250.3 ± 1.1	234.8 ± 4.6
	TM	-	9.0 ± 0.6	28.5 ± 1.4	40.1 ± 3.0	45.1 ± 2.7	76.4 ± 1.5	101.6 ± 3.5	97.6 ± 2.4	140.9 ± 2.0	169.6 ± 0.7	205.0 ± 2.9	207.8 ± 12.2	248.2 ± 1.2	241.5 ± 7.1	225.9 ± 17.1
	Sig.	-	*	NS	NS	**	NS	NS	NS	NS	*	NS	NS	NS	NS	NS
WI	C	-	21.5 ± 0.6	47.9 ± 1.4	83.6 ± 2.7	133.4 ± 3.4	219.3 ± 9.7	231.3 ± 6.5	242.0 ± 7.4	298.0 ± 14.4	410.0 ± 15.7	410.3 ± 21.2	301.3 ± 19.4	409.8 ± 7.8	411.7 ± 18.5	404.0 ± 13.2
	TM	-	14.7 ± 1.3	29.4 ± 1.0	55.2 ± 2.3	92.0 ± 5.1	113.0 ± 6.5	161.2 ± 5.0	165.9 ± 3.9	220.4 ± 3.2	287.6 ± 10.9	328.4 ± 2.3	266.4 ± 8.6	349.0 ± 10.6	330.1 ± 6.0	361.0 ± 20.8
	Sig.	-	**	**	**	**	NS	NS	NS	NS	NS	**	NS	NS	NS	NS
FCR	C	-	1.8 ± 0.1	1.8 ± 0.1	1.8 ± 0.1	2.2 ± 0.2	2.1 ± 0.1	2.3 ± 0.1	2.2 ± 0.1	2.5 ± 0.1	2.8 ± 0.0	3.4 ± 0.0	5.1 ± 0.3	3.1 ± 0.2	3.7 ± 0.2	5.0 ± 0.1
	TM	-	1.7 ± 0.1	2.5 ± 0.1	2.5 ± 0.1	2.3 ± 0.1	2.3 ± 0.1	2.2 ± 0.0	2.2 ± 0.1	2.4 ± 0.1	2.9 ± 0.1	4.0 ± 0.5	3.8 ± 0.4	2.9 ± 0.1	3.9 ± 0.3	4.9 ± 0.3
	Sig.	-	NS	**	**	NS	NS	NS	NS	NS	NS	NS	NS	NS	NS	NS

NS, * or **: non-significant or significant at *p* < 0.05 or 0.01, respectively; SEM: standard error of the mean, P: parameter, G: group, C: control group, TM: tenebrio group, Sig.: significance, w: week, d: days, LW: live weight, BWG: body weight gain, FI: feed intake, WI: water intake and FCR: feed conversion ratio; ^1^ LW was punctually measured the last day of each week, i.e., on days 1, 8, 15, 22, 29, 36, 43, 50, 57, 64, 71, 78, 85 and 95.In general, during the whole experimental period, BWG was significantly higher (*p* < 0.01) for chickens of group C. Furthermore, FI and WI were also significantly higher (*p* < 0.05 and *p* < 0.01, respectively) in group C. However, in both LW and FCR there were no significant differences between the two groups, albeit the LW deviation was higher in the TM group (Figure 1).

### 3.2. Growth and Feed Intake Increases

Chicken growth was highest in the early stages of life (Figure 2). The BWG between successive periods was significantly different (*p* < 0.01) throughout the trial. Analyzing each group separately, the greatest BWG corresponded to the period between days 8 and 15, the greater increase being for group C than group TM (128.45% vs. 100.36%, respectively). After 36 days, coinciding with having had less *T. molitor* content in the rations, the weight increase was lower than in the previous stages, though the increase was higher for the TM group.

The greatest differences in BWG for a specific period between the two groups were found in the mid-term days of the trial (Figure 3). Significant results were found from 15 days of life to the end of the trial (*p* < 0.01), during which time group C reached higher weights. The greatest difference in BWG between the two groups occurred at 36 days, when the weight of group C exceeded the weight of group TM by 32.21%. At the end of the trial, the differences between groups decreased.

Significant differences (*p* < 0.01) in the FI increase between the two groups (Figure 4) occurred from the 4th to 10th weeks. The greatest difference between the two groups occurred in the 4th week, group C being the one that exceeded the TM group by 36.20%. From day 29, i.e., when they changed from the F1 to the F2 diet, differences were still significative but diminished.

## 4. Discussion

There are relatively few studies involving *Tenebrio molitor* meal for slow-growing chickens for 95 days, as the growth cycle is usually shorter (approximately 60 days) [27,28]. The results obtained in this work are partially in agreement with those found by Benzertiha et al. [29], who determined that the addition of insect meal during the first 35 days of life of chickens improved their BWG and FI, but increased their FCR.

During the period of F1 intake, there was a reduction in FI and a lower final LW. In the compositions of the diets, non-important differences were observed, except for a lower fat content in the TM group. However, the amino acid contents were very similar in both groups, which did not explain the obtained differences. Therefore, it could be considered that the proportion of *Tenebrio molitor* used during the first days of life may affect the total digestibility of the diet.

During the feeding period of the second and third diets (F2 and F3), no differences were found in growth parameters except for BWG in F2. These results coincide with those exposed by Biasato et al. [13], who pointed out that the inclusion of insect meal did not negatively affect chicken performance, supporting the feasibility of its inclusion [30]. Between 30 and 62 days, Bovera et al. [17] found no differences in FI in chickens when meal from this same insect species was used to replace soybean meal in isoproteic and isoenergetic diets, which is in line with our results regarding FI. Furthermore, the results of this work also coincided with those of Biasato et al. [13], who claimed that *T. molitor* meal inclusion did not affect productive performance of chickens by either the 43rd or the 97th day of life.

In the middle and final periods of the trial, no differences in LW were found, contrary to the results of Biasato et al. [19], who found that chicken LW improved when *T. molitor* larval meal was included. Throughout the entire trial, no differences in BWG or FI were found, agreeing with the results of Leiber et al. [6], in which FI and BWG were equal for both chickens that received insect meal and those that did not over a period of 7 to 82 days. The overall FCR of the trial showed no differences between groups. This is in agreement with the results described by Biasato et al. [13], in which the inclusion of *T. molitor* meal in the period from 43 to 97 days did not affect the FCR. During the first 4 weeks, the FCR exhibited differences between the two groups. There were higher values for the TM group, which is consistent with the results of Benzertiha et al. [29], whose work indicated that the inclusion of *Tenebrio molitor* negatively affected the FCR during the first 35 days. According to Bovera et al. [27], FCR improved in chickens fed *Tenebrio molitor* from 30 to 62 days; in the present study, no differences in F2 were found. Even so, better results in both groups were recorded than those obtained by other authors [27].

The results of this work demonstrate that, despite the fact that during the first month the performance of the chickens in the TM group was lower, at the end of the growth cycle it was equivalent to that of the C group. Therefore, starting from smaller chickens in the first month, the substitution of soybean meal by *Tenebrio molitor* meal improves the productive performance of the animals in the later stages [29]. Consequently, this insect meal is suggested as a feasible alternative to replace soybean in poultry diets [7]. This is also supported by the conclusions elucidated by other authors [31], who obtained better performance in chickens that were fed insect meal. For Leiber et al. [6], the LW of chickens fed insect meal was also higher. In other experiments conducted with fattening poultry, increasing the inclusion of *T. molitor* in quail diets could improve LW and FCR [24].

High amounts of insect meal in early life stages can be detrimental to chickens, as it reduces FI and LW [32,33]. Thus, some authors have obtained better results by making partial substitutions, suggesting that the optimal ratio of soybean meal to insect meal substitution depends on the age of the animals [34]. For Biasato et al. [18], progressively increasing *Tenebrio molitor* content in the diet during the first 25 days impaired FI and FCR of chickens. This conclusion is supported by other studies, which concluded that the inclusion of *T. molitor* meal in chickens diets can improve LW and FI, albeit it could negatively affect FCR if inclusion levels are not adequate [19].

The alternative feed tested could partially replace soybean feeds in chicken diets, but full substitution would not be appropriate [6]. In this context, Ravindran and Blair [35] noted that the chitin contained in the insect exoskeleton is hard to digest by chickens. This situation is even more acute in the early stages of chickens’ lives, in which their digestive system is still in an immature state, and in addition, there is a tendency to increase the insect meal intake due to the high protein requirements at this stage of growth. Therefore, in the first month of life, soybean should continue to be part of the diets of chickens, its substitution being feasible 5th week of life onwards. This fact reaffirms the idea of Olkowski [36], who tried to replace soybean meal with lupin meal in a chicken diet, concluding that it could be carried out from the end of 4 weeks onwards, because the application of high levels of yellow lupin seed in the initial rations produced a negative effect on bird performance. The results of the present study also show that the initial differences in growth between the two groups decreased as the trial progressed, coinciding with a lower inclusion of insect meal in the diet.

## 5. Conclusions

In this work, we tried to assess the effect of replacing soybean meal as the main protein intake with *Tenebrio molitor* larvae meal in isoproteic and isoenergetic diets for slow-growing chickens. The results suggest that such a high percentage of insect meal should not be included in the first days of life of the animals (up to 29 days). Therefore, a partial substitution would be more appropriate during this first period, since after it, the chickens improved notably their development and performance, reaching at the end of the cycle (95 days) values similar to those of chickens fed soybean meal, despite starting from animals with less development during the first month. Under these conditions, *T. molitor* larval meal can be an alternative protein source to include in the diets of slow-growing chickens.

## Figures and Tables

**Figure 1 vetsci-09-00131-f001:**
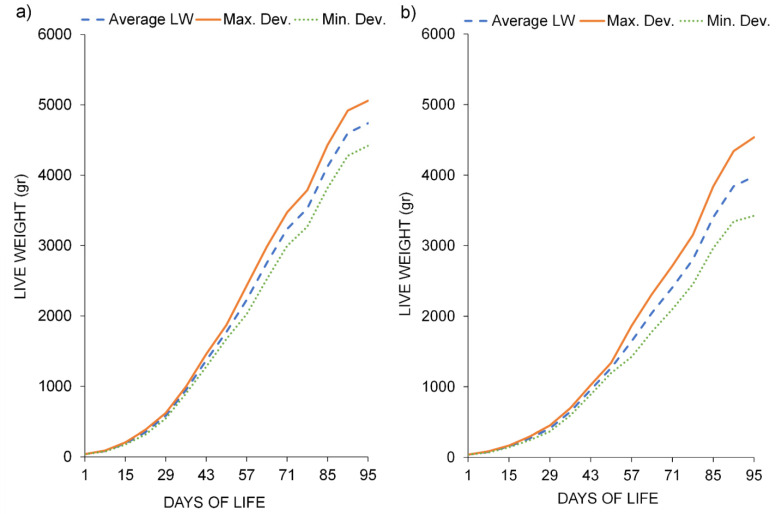
LW curves for the two studied groups (C, Control group and TM, Tenebrio group) throughout the growing period (95 days). (**a**) Group C and (**b**) group TM. LW: live weight, Max Dev.: maximum live weight deviation and Min. Dev.: minimum live weight deviation.

**Figure 2 vetsci-09-00131-f002:**
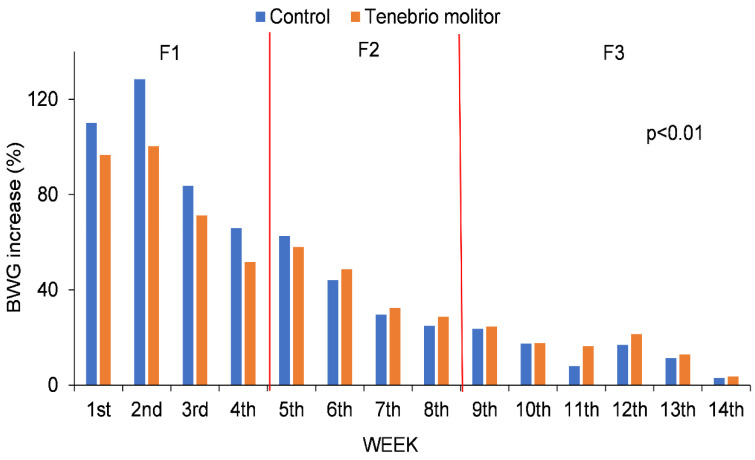
BWG (Body Weight Gain) percentage increase in each studied group (C and TM) between two consecutive weeks throughout the whole study period.

**Figure 3 vetsci-09-00131-f003:**
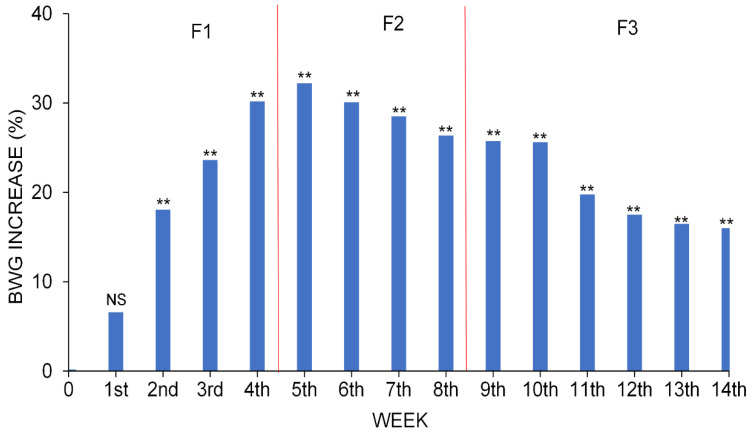
BWG percentage increase in the C group over the TM group throughout the whole study period. NS or **: non-significant or significant at *p* < 0.01, respectively.

**Figure 4 vetsci-09-00131-f004:**
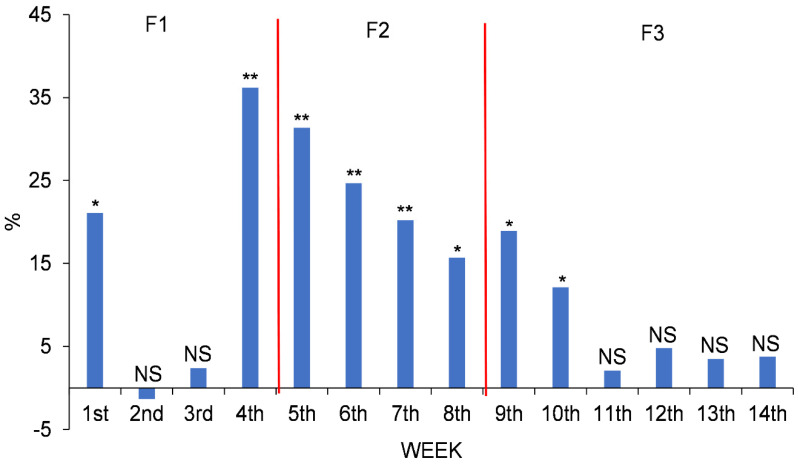
FI percentage increase in the C group over the TM group throughout the whole study period. NS, * or **: non-significant or significant at *p* < 0.05 or 0.01, respectively.

**Table 1 vetsci-09-00131-t001:** Compositions of the experimental diets for the two studied groups of chickens expressed as percentages of raw matter.

Raw Matter	F1 (1–29 d)		F2 (29–57 d)		F3 (57–95 d)	
	C	TM	C	TM	C	TM
Corn	37.0	10.0	48.0	9.0	31.0	18.0
Wheat	11.0	0.0	12.0	16.8	37.0	33.5
Barley	0.0	21.0	0.0	13.2	0.0	0.0
Soybean meal	29.0	0.0	20.0	0.0	12.0	0.0
*T. molitor* meal	0.0	22..3	0.0	13.0	0.0	9.2
Sunflower	19.0	0.0	16.0	9.0	16.0	9.0
Oats	0.0	42.7	0.0	35.0	0.0	26.3
Vitamin-mineral premix	4.0	4.0	4.0	4.0	4.0	4.0

C: control group, TM: *T. molitor* group and d: day.

**Table 2 vetsci-09-00131-t002:** Chemical/nutritional characteristics of the experimental diets for the two studied groups of chickens.

Parameter	F1 (1–29 d)		F2 (29–57 d)		F3 (57–95 d)		*T. molitor* Meal
	C	TM	C	TM	C	TM	
%Moisture	7.6	7.9	8.9	8.9	8.6	8.5	6.6
%Ash	6.5	6.4	6.4	6.0	5.8	5.9	4.2
%Crude fat	12.3	9.9	9.5	9.1	10.7	9.7	21.2
%Crude fiber	6.1	7.9	5.0	8.5	6.1	6.9	8.3
%Starch	35.0	38.1	41.2	37.4	42.3	40.9	2.3
%Crude protein	19.7	19.3	15.6	15.5	14.8	14.2	54.9
ME (Kcal/kgDM)	2802.8	2898.9	2900.0	2900.7	2950.8	2964.7	5020.0
Mean dietary Crude Protein (%)	19.5		15.6		14.5		54.9
Mean dietary ME (Kcal/kgDM)	2850.8		2900.3		2957.8		5020.0
%Ca	1.24	1.94	1.21	1.63	1.20	1.49	3.6
%P	0.40	2.10	0.36	1.41	0.34	1.08	7.4
%SFA	1.6	2.2	1.2	1.6	1.4	1.5	5.0
%MUFA	3.0	3.7	2.3	2.8	2.4	2.8	8.4
%PUFA	7.8	4.1	6.0	4.7	7	5.4	7.9
%n-3 PUFA	0.15	0.14	0.11	0.09	0.10	0.08	0.25
%n-6 PUFA	7.6	3.9	5.9	4.7	6.8	5.3	7.6
%Trans FA (C18:1T + C18:2T + C13:3T)	<0.05	<0.05	<0.05	<0.05	<0.05	<0.05	<0.05
4-Hidroxiproline (mg/100 gDM)	59.0	24.0	46.0	24.8	39.9	23.7	72.0
Aspartic acid + Asparagine (g/100 gDM)	2.54	1.83	1.92	0.84	1.20	0.86	4.5
Glutamic acid + Glutamine (g/100 gDM)	5.22	4.13	4.45	2.83	4.12	2.86	6.4
Alanine (g/100 gDM)	0.91	1.40	0.90	0.92	0.95	0.99	5.0
Arginine (g/100 gDM)	1.28	0.91	1.21	0.70	0.77	0.80	2.74
Cysteine (g/100 gDM)	<20	<20	<20	<20	<20	<20	<20
Cystine (g/100 gDM)	0.18	0.13	0.15	0.11	0.12	0.11	0.30
Phenylalanine (g/100 gDM)	0.81	0.59	0.68	0.55	0.57	0.45	2.17
Glycine (g/100 gDM)	1.02	0.61	0.88	0.45	0.40	0.67	2.72
Histidine (g/100 gDM)	0.76	0.57	0.65	0.95	0.76	0.48	1.51
Isoleucine (g/100 gDM)	0.68	0.44	0.53	0.58	0.40	0.36	2.39
Leucine (g/100 gDM)	1.18	0.94	1.09	1.04	0.94	0.76	4.9
Lysine (g/100 gDM)	1.44	1.23	1.28	0.97	1.01	0.69	2.80
Methionine (g/100 gDM)	0.29	0.18	0.20	0.17	0.19	0.17	0.47
Proline (g/100 gDM)	0.91	1.00	1.45	1.03	1.03	1.16	2.74
Serine (g/100 gDM)	0.90	0.88	0.90	0.80	0.76	0.59	2.25
Tyrosine (g/100 gDM)	0.58	0.69	0.46	0.81	0.34	0.80	4.1
Threonine (g/100 gDM)	0.79	0.64	0.58	0.45	0.50	0.65	1.89
Tryptophan (g/100 gDM)	<20	<20	<20	<20	<20	<20	<20
Valine (g/100 gDM)	0.65	0.61	0.53	0.65	0.46	0.41	4.7
Hydrolyze protein (g/100 gDM)	20.22	16.83	17.80	13.94	14.56	12.87	52

C: control group, TM: *T. molitor* group, d: day, ME: metabolizable energy, Ca: calcium, P: phosphorous, SFA: saturated fatty acids, MUFA: monounsaturated fatty acids, PUFA: polyunsaturated fatty acids, n-3 PUFA: omega-3 polyunsaturated fatty acids, n-6 PUFA: omega-6 polyunsaturated fatty acids, Trans FA: trans fatty acids and DM: dry matter.

**Table 3 vetsci-09-00131-t003:** Total production performance parameters of the two studied chicken groups (C and TM) for the three different diets (mean ± SEM).

Parameter	F1 (1–29 d)			F2 (29–57 d)			F3 (57–95 d)				Total (1–95 d)	
	C	TM	Sig.	C	TM	Sig.	C	TM	Sig.	C	TM	Sig.
LW ^1^	583.6 ± 13.6	407.5 ± 14.8	**	2229.0 ± 45.3	1643.3 ± 49.2	NS	4739.0 ± 73.5	3981.5 ± 124.4	NS	4739.0 ± 73.5	3981.5 ± 124.4	NS
BWG	19.4 ± 0.5	13.1 ± 0.5	**	60.2 ± 1.9	44.3 ± 2.1	*	66.0 ± 1.4	63.8 ± 1.2	NS	49.99 ± 0.78	43.8 ± 1.2	**
FI	1059.4 ± 41.6	858.8 ± 47.0	**	3749.9 ± 71.7	2915.7 ± 78.6	NS	8865.0 ± 41.1	8182.8 ± 193.8	NS	13,674.30 ± 198.56	11,957.4 ± 282.6	*
WI	2004.3 ± 48.8	1339.4 ± 43.5	**	6934.2 ± 178.1	4623.6 ± 118.5	NS	14,813.5 ± 618.5	12,014.1 ± 54.1	NS	23,751.88 ± 601.13	17,977.1 ± 427.6	**
FCR	2.0 ± 0.1	2.3 ± 0.1	**	2.3 ± 0.0	2.4 ± 0.1	NS	3.5 ± 0.0	3.5 ± 0.1	NS	2.9 ± 0.1	3.0 ± 0.1	NS

NS, * or **: non-significant or significant at *p* < 0.05 or 0.01, respectively; SEM: standard error of the mean, C: control group, TM: tenebrio group, Sig.: significance, d: days, LW: live weight, BWG: body weight gain, FI: feed intake, WI: water intake and FCR: feed conversion ratio; ^1^ LW was punctually measured on the last day of the week, coinciding with the end of each period, i.e., on days 29, 57 and 95.

## Data Availability

The data presented in this study are available on request from the corresponding author. The study did not include humans.

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
