# Peer review of "Performance of Slow-Growing Chickens Fed with Tenebrio molitor Larval Meal as a Full Replacement for Soybean Meal"

_vetsci, 2022, doi:10.3390/vetsci9030131_

Round 1

Reviewer 1 Report

Manuscript ID: VetSci-1621289

Type of manuscript: Research paper

Title: Performance of slow-growing chickens fed with Tenebrio molitor larval meal as a full replacement for soybean meal

Comments to the authors:

The manuscript aimed to highlight and evaluate the performance of slow-growing chickens fed with T. molitor larvae meal as a complete replacement for soybean meal.

According to the reviewer, the use of T. molitor as an alternative protein source to replace soybean meal in slow-growing chickens is an interesting study. Moreover, it is a trending topic in animal feeding and production to improve its sustainability and product quality for the foreseeable future. The manuscript is well structured and deals appropriately with the primary purpose of the study. However, the reviewer has some comments that may help improve the understanding of the manuscript.

Reviewer recommendation: Accepted (with minor revision)

Comments:

General: At the end of the trial, chickens were slaughtered? If yes, then why the authors did not consider displaying performances and carcass traits together? It could be easier to interpret data.

Line 15-27: The reviewer suggests the authors explain the three different concentrated diets (F1, F2, and F3) in the abstract before using abbreviations

Line 39: Reviewer suggests the authors to cite an appropriate reference as the cited reference deals with poultry species only

Line 40: The cited reference does not correspond to the sentence

Table 1: Corrector? What does it mean? Is it vitamin-mineral premix, salt, etc…?

Table 2: Please replace “Ashes” with “Ash”. How about calcium and phosphorous content in the experimental diets?

According to the reviewer: introduction, statistical analysis, results, and discussion are at a satisfactory level

Author Response

Consulte el archivo adjunto

Reviewer 2 Report

This manuscript is well written and easy to read. It is an interesting and relevant topic for the animal feeding sector. The description of the experiment is clear in regards to the objectives of the project.  Please find below some questions / comments but I suggest to accept this article for publication.

L9-L10 : A natural behaviour of chicken is to consume insect, but do you think this is relevant when taking about insect meal that will be incorporated into a pellet ?

L69 : It would be great to better justify why you focused on Tenebrio Molitor and not on other insect species (such as Hermetia illucens which is also often used in feeding trials).

L102 and 111 : please precise if diets were pelleted and if feed and water were provided ad libitum.

L 122 : iso-proteically : does it means iso-nitrogen ? iso-amino acids or iso-digestible amino acids ? Which criterion did you used for diet formulation ?

Table 1 : What is the Corrector (4%) ? I guess minerals, vitamins…but this should be mentioned and the composition should be detailed.

Table 2 : A large number of analyses have been carried out. This should be better used in the discussion to argue or make hypotheses about differences observed between diets.

L 150 Statistical analyses : please check the conditions of validity before using these tests (e. g. normality of ditribution...)

L 165 : What about the mortality ?

Figure 1 : Please use different signs to better differenciate curves

Figure 2 : Please add « BWG increase (%) » in the y axis. Could the x axis be prensented as weeks similarly to Figure 4 for better consistency ? The presentation is a bit confusing. Which conditions are significantly different from others, this is not very clear.

Figure 3 : Please add « BWG increase (%) » in the y axis. Could the x axis be prensented as weeks similarly to Figure 4 for better consistency ? Delete * significant at p<0.05 in the legend as it does not appear in the figure.

L 294 :

Did you measure the chitin level in your diets to compare with others studies and the levels that chickens « are able to digest » ?

What could be others hypothesis to explain the results observed in the starter period ? If diets were balanced on the crude protein level, do you think the level and the balance of digestible amino acids could also be an explanation ?

The dry matter of C and TM diets are quite similar so how could you explain the effect of the diet on water intake ?  Did you see any positive consequence of that (litter quality) ?

Reviewer 3 Report

The Materials & Methods section provides insufficient information:

- the Authors have not provided the details regarding the  performance data collection.

- why did you analyze the fatty acid profile of the diets when it was not determined in chicken meat ?.

 - Do these chickens had free access to feed and water ?.

- Were the animals weighed individually within the experimental groups and also individually within the respective replicates? Or were the animals within the replicates weighed as a group?

- why do you mention that the chickens were slaughtered when you determined only the parameters of live birds?

- The results of the chemical composition of the larval meal are needed !!.

- Detailed information is needed on how the ME content of diets was estimated. Contrary to the authors' findings, I believe that the starter diets were not isoenergetic (2,803 vs. 2,899 Kcal/kg).

Finally, there is no mention of the bird mortality results.

Round 2

Reviewer 3 Report

no comments